# Experiences of social support and the role of engagement in a digital educational support group for adolescent mothers' health in the Dominican Republic

**Alana R. Lopez** [1,2]*, **Elizabeth Haight**[3], **Linda Guijosa**[3], **Briana Williams** [2,4], **Jennifer Unger**[2], **Luz Messina**[5], **Mina Halpern**[5], **Samantha Stonbraker**[6], **Keshet Ronen**[2]

**1** Herbert Wertheim School of Public Health and Human Longevity Science, University of California San Diego, San Diego, California, United States of America, **2** Department of Global Health, University of Washington, Seattle, Washington, United States of America, **3** Planned Parenthood of the Great Northwest, Hawai'i, Alaska, Indiana, Kentucky: Global Programs, Seattle, Washington, United States of America, **4** Department of International Health and Sustainable Development, Tulane University, New Orleans, Louisiana, United States of America, **5** Clínica de Familia La Romana, La Romana, Dominican Republic, **6** College of Nursing, University of Colorado Anschutz Medical Campus, Aurora, Colorado, United States of America

* arl011@ucsd.edu

## Abstract

In 2021, nearly 66 of every 1,000 adolescent girls ages 15–19 in the Dominican Republic gave birth. Adolescent mothers face health disparities including increased risk for rapid repeat pregnancy and lower breastfeeding rates. Mobile health (mHealth) is a growing approach for reaching adolescents. FAMA (*Fortaleciendo la Autodeterminación de Madres Adolescentes)* was a 12-week moderated digital education support group with adolescent mothers in the Dominican Republic, associated with improvements in health knowledge and contraceptive uptake. This study explores the FAMA intervention's mechanism of action through a mixed-methods secondary data analysis using WhatsApp messages and post-intervention interviews to characterize experiences of social support and patterns of intervention engagement. We assessed associations between multiple measures of engagement and intervention outcomes. Linear or Poisson regression was used to evaluate association with health knowledge, and social support. or contraceptive uptake, respectively, selected based on type and distribution of each outcome variable. Models adjusted for key confounders. Findings indicate FAMA was largely used to exchange companionship and informational support. We found a significant positive association between engagement as measured by acknowledging intervention messages and improved health knowledge (adjusted coefficient: 2.84, CI: 0.83-4.84, p= 0.01). In contrast, we found a negative association between engagement as measured by social support exchange and improved health knowledge (adjusted coefficient: -5.78, CI: -10.42- -1.00, p= 0.02), suggesting that interactions focused on support may not reinforce informational content as directly as other forms of engagement. Our findings suggest that engagement with FAMA was associated with increases in knowledge and a close reading of message content is most beneficial for knowledge gain. This analysis enhances understanding of

**Data availability statement:** De-identified survey data used for this analysis are within the manuscript's Supporting Information files. Qualitative and WhatsApp data cannot be shared publicly as they contain personally identifiable information. These data are available from Clinica de Familia La Romana (info@ clinicadefamilia.org.do) for researchers who meet the criteria for access to confidential data.

**Funding:** This project was supported by Grand Challenges Canada, award number ST-POC-1808-17557 (to EH, LM, MH) and Grants K18MH122978 and P30AI027757 from NIH (to KR). Grand Challenges Canada is funded by the Government of Canada and is dedicated to supporting Bold Ideas with Big Impact. The funders had no role in study design, data collection and analysis, decision to publish, or preparation of the manuscript.

**Competing interests:** The authors have declared that no competing interests exist.

user engagement with group mHealth interventions and contributes new approaches to measure engagement, accounting for different engagement styles participants may have. Future digital interventions may leverage our findings to design interventions that encourage beneficial engagement types.

## Author summary

Birth rates among adolescent mothers in the Dominican Republic are high, and these mothers face many health challenges, including higher risks for depression and lower rates of breastfeeding. To address some of these disparities, a 12-week digital education support group called FAMA (Fortaleciendo la Autodeterminación de Madres Adolescentes) was developed. This mobile health program aimed to improve health knowledge and contraceptive uptake among adolescent mothers in the Dominican Republic. In this analysis, we explored how participants engaged with the program and the relationship between their engagement and health outcomes. Our findings showed that overall engagement in the group was positively associated with improved health knowledge. However, the different styles of engagement we examined showed distinctive relationships with changing health knowledge. Our findings contribute to the understanding of mobile Health intervention engagement and suggest that future digital group interventions should consider diverse engagement styles and the specific needs of adolescent mothers.

## Introduction

In low and middle-income countries, approximately 21 million adolescents ages 15–19 become pregnant each year [1]. Of these, nearly 20% of adolescent pregnancies resulted in the birth of a second or high-order child [2]. Parenting and rapid repeat pregnancy—a short interval before subsequent pregnancy—in adolescence pose health risks as well as long-term financial, emotional, and social risks for the parent and infant [3]. Adolescent mothers are more likely than older mothers to live in poverty and experience disproportionately low educational attainment [4]. Additionally, adolescent mothers are at higher risk of depression [5], are less likely to breastfeed their infants [6], have increased risk of infant developmental delay [5], and increased infant mortality [5]. Younger mothers, particularly those in low and middle income countries, have been found to be at higher risk for shorter intervals between births [7]. Research shows these shorter intervals are associated with increased risks of preterm birth, low birthweight, infant mortality, and maternal death [8]. Improving health indicators for adolescent mothers such as contraceptive use is, therefore, a critical priority.

Among countries with high adolescent birth rates, the Dominican Republic (DR) stands out as a particularly urgent case. The DR has the highest adolescent fertility rates in Latin America and the Caribbean and is in the 87th percentile in the world [9]. In 2021, nearly 66 of every 1,000 adolescent girls ages 15–19 in the DR gave birth, compared to the global average of 42.5 per 1,000 adolescent girls [10]. Recent research suggests that these values are likely to be an underestimation of the total adolescent fertility rate in the DR, as they do not include adolescents below age 15 [11]. In the DR, approximately 27% of 15–19-year-olds must overcome numerous barriers to access contraceptives [12], and young Dominicans demonstrate limited sexual and reproductive health knowledge [13].

Given the numerous barriers and risks faced by adolescent mothers, social support is key for this group [14], whose developmental stage is characterized by increased reliance on and trust of peers [15]. Adequate social support, particularly peer support, has been associated with improved knowledge about child development, up to a 30% reduction in the risk of not breastfeeding [16,17], and 70% higher odds of using modern contraceptives [18]. Social support can take a variety of forms, including companionship (social interaction to provide mutual enjoyment [19]), emotional (caring, empathy, love, and trust [20,21]), informational (information provided during a stressful time, which assists in problem-solving [20,21]), and instrumental (tangible support such as goods or services [20,21]). Support groups for new mothers can provide social support by facilitating peer relationships and normalization of shared experiences [22].

Mobile health (mHealth), the use of mobile phones to improve health, is an increasingly popular approach to deliver information and provide support for patients, particularly adolescents and young adults [23–29]. Globally, mobile phone use is very high, especially among young people [23]. This presents an opportunity to reach beyond clinic settings to potentially overcome barriers to in-person care experienced by adolescents [30,31]. mHealth interventions have been shown to be highly acceptable to adolescents [32,33] and mothers [34,35], and to influence postpartum contraceptives uptake [36,37], improve reproductive health knowledge [38], and increase exclusive breastfeeding [37]. Although mHealth interventions show promise for engaging young people, challenges sustaining user engagement have been noted [39,40], highlighting a gap in understanding the key forms of user engagement and ways to optimize it. This study aims to address this gap by identifying the ways adolescent mothers engage with a digital health intervention, and exploring which forms of engagement may be more closely involved in observed changes in participants.

Our team previously developed and evaluated a novel 12-week moderated WhatsApp educational support group with adolescent mothers in the DR which aimed to improve maternal and infant health through increased social support and access to health information [41,42]. To explore the potential mechanism of action of this educational support group, we present an analysis of participant self-reported experiences and observed WhatsApp messaging to determine the domains of social support exchanged in the intervention, define patterns of participant engagement with the digital educational support group, and identify associations between patterns of engagement and selected outcomes (social support, health knowledge and postpartum health behaviors). This digital educational support group's hypothesized mechanism of action (Fig 1) is largely influenced by Social Support Theory [43] as participants received emotional, information, instrumental, and companionship support as part of the digital support group intervention. We hypothesized that participants' engagement with daily informational support messaging provided by the intervention leads to improved health knowledge, while broader social support provided by a moderated peer group chat leads to higher perceived social support. We hypothesized that improved health knowledge and other social support leads to uptake of contraceptives.

## Materials and methods

We conducted a secondary analysis using data from the Strengthening Self-Determination of Adolescent Mothers (*Fortaleciendo la Autodeterminación de Madres Adolescentes* (FAMA)) intervention, conducted at *Módulo Anexo Materno Infantil (MAMI)*, an adolescent health facility run by the non-profit Clínica de Familia La Romana, in collaboration with the public Hospital Francisco A. Gonzalvo, in La Romana, Dominican Republic. Briefly, the intervention began with an in-person group meeting to build connection among mothers. Mothers then

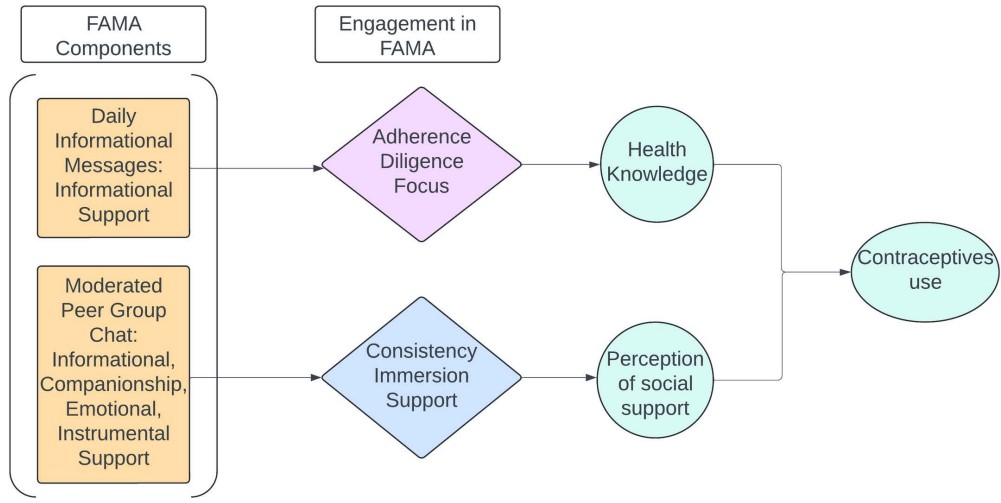

**Fig 1. FAMA proposed mechanism of action.**

participated in one of three 12-week long, 16–22-person WhatsApp groups, in which daily informational messages were sent by one of two moderators regarding health-related themes, and participants were free to send WhatsApp messages to the group at any time. Daily informational messages were related to the following topics, identified in preliminary work, (two weeks each): family planning, baby illnesses, growth and development stages, feeding baby, understanding baby, and breastfeeding. Participants were expected to participate daily by sending the "emoji of the day" to confirm they had received and read the information for the day. The selected emoji of the day was included at the end of each daily message along with a reminder to participants to send it. Moderators also sent forum questions every two weeks which aimed to spark discussion and engagement in the chat group related to that week's topic. Participants were encouraged, but not required, to respond to forum questions in the group. In addition, participants were encouraged to ask questions, seek support, and engage with each other at any time in relation to the topics discussed, the health of their babies, and their own health. Following the receipt of several messages at odd hours, participants were told that moderators were available to answer questions and participate in groups only from 8 A.M. to 5 P.M. on weekdays. After the WhatsApp intervention, a closing in-person group meeting was held where a subset of participants were asked to participate in individual interviews about their experiences with FAMA. This digital educational support group was associated with improved health knowledge, increased contraceptive use, and increased well-baby visit attendance [41].The current study uses a concurrent mixed methods approach. We quantified FAMA group messages and integrated quantitative messaging data with qualitative interview data at the stage of analysis.

## Ethical approval and consent

Ethical approval for this study was obtained from the Consejo Nacional de Bioética en Salud (CONABIOS), the ethical review committee in the DR and the University of Washington Institutional Review Board. All participants provided written informed consent to participate in the intervention and allow study team members access to their medical records. Verbal informed consent was later obtained for those participants who agreed to be interviewed after completing the intervention. In accordance with Dominican law 136–03, adolescents under 18 who were pregnant and/or have a child are considered emancipated minors and can provide

informed consent for study participation without parental permission. To protect confidentiality, all participant data was identified using a study identification number, unlinked to any participant identifiers

## Study participants

Participants were recruited by study staff in the waiting room at MAMI through direct outreach. Study staff approached individuals identified as eligible by the clinic receptionist and provided a detailed study overview. Interested participants were then consented and enrolled. Participants were eligible for study participation if they were 14–19 years old, had at least one child, had given birth in the last six months, spoke Spanish, lived in La Romana, and received health services at MAMI. Care at MAMI included access to contraceptive methods. The final study sample included 58 participants.

## Data collection

**Post-intervention interviews.** A convenience sample of participants were approached during the final in-person meeting of the FAMA intervention and asked to participate in an individual interview. Structured interviews were conducted in Spanish by study staff using a guide that was created by the study team with questions and prompts to explore participants' perspectives towards the acceptability of the intervention, experiences participating in the study, and to obtain feedback regarding how to improve FAMA for future iterations. Interview methods were previously described in detail [41].

**WhatsApp chats.** WhatsApp chats from the three FAMA groups were exported from a study team member's device as text files; messages and timestamps were imported into Microsoft Excel for content analysis. Audio messages were individually transcribed. Photo and video message content was excluded from this analysis as it was not saved in data export. "Sticker" media messages were exported, and their content was summarized as text. To ensure participant confidentiality in handling exported WhatsApp messages, each participant was assigned a study identification number, and participant identifiers such as phone numbers and names were removed from the data.

**Surveys.** Paper surveys were administered in-person during group meetings at the start and end of the

12- week group period. Surveys collected information on demographic characteristics, health knowledge, and social support. The health knowledge survey was an 11-item questionnaire created for the FAMA study based on the key themes and informational messages provided in the intervention including infant nutrition, family planning, breastfeeding, infant illnesses, infant growth and development, and other general infant care (possible score range 0–11).

The social support survey was based on the NIH Toolbox Social Relations scales, using measures from the Patient Reported Outcomes Measurement Information System (PROMIS) tools which have been validated in Spanish and assess emotional support, informational support, social isolation, means of support and companionship [44,45]. Although these scales have been individually validated, the present study combined scaled scores across all five scales to give a more complete understanding of each participant's experience of social support. Scores for each subscale were scored using the T-score Metric wherein a score of 50 is the mean of a reference population, and scores range from 20-80 [44].

**Clinical data.** Clinical data for each participant was gathered by study team members by review of medical files at the study facility. Information on participant use of contraceptives (oral, injectable, and other long-acting contraceptives) was recorded for each participant throughout the intervention period.

## Data analysis

**Quantitative analysis.** Quantitative data sources included pre- and post-intervention surveys, contraceptive use data abstracted from clinic records, and quantitative summaries of WhatsApp messaging. WhatsApp chats were quantitatively summarized as indicators of intervention engagement, described below. Qualitative coding and categorization of messages allowed for quantitative analysis of participant engagement through message type and content.

We developed an Engagement Index as a summary measure of participant engagement in FAMA, allowing for multiple styles of user engagement. The development of this measure of engagement was necessary as WhatsApp does not provide metadata such as time spent in a chat, or time spent reading messages. Our Engagement Index was adapted based on [46]:

$$EI = \sum \left( A_i + D_i + C_i + S_i + I_i + F_i \right)$$

where EI is Engagement Index, A is Adherence Index, D is Diligence Index, C is Consistency Index, S is Support Index, I is Immersion Index, and F is Focus Index. The *Adherence Index* was calculated as the proportion of emojis of the day sent, out of the total possible days in which informational messages were sent. This was the only compulsory response participants were expected to send to the group each day to confirm receipt of the informational message sent by the moderator. The *Diligence Index* was calculated as the proportion of group forum questions answered over the 12-week intervention period. Forum question responses were identified during qualitative coding of chats. The *Consistency Index* was calculated as the proportion of intervention days a participant sent any message containing text. This index demonstrates consistent interaction with the group beyond acknowledging the informational message with the emoji of the day. The *Support Index* was calculated as the proportion of total messages in which each participant demonstrated an exchange (either seeking or offering) of social support (companionship, emotional, instrumental, informational), based on qualitative coding of WhatsApp chats. The *Immersion Index* was calculated as the participants' percentile rank of the total number of messages sent throughout the 12-week period. Each participant's rank was calculated among the members of their WhatsApp group to allow for differences in group dynamics. The *Focus Index* was calculated as the percent of messages sent by each participant which were identified as responses to that day's topic or were related to one of the six content focus areas of the intervention. All indices were calculated separately for each participant. Each index has a possible range of scores of 0–1 for each participant. The total EI range is 0–6.

We examined the association between messaging engagement and outcomes collected at baseline and endline. Association between the EI and health knowledge and social support scores at study exit was determined by linear regression. This analysis was used as it is appropriate for use with continuous outcome variables. Association between EI and contraceptive use at study exit was determined by Poisson regression with robust standard errors. This analysis was used as it allows for the approximation of relative risk for a common binary outcome, while accounting for overdispersion. We conducted univariable and multivariable analyses adjusted for *a priori* defined confounding factors: participant age, whether the participant had legal documentation status (i.e., had documentation of birth necessary to enroll in university and participate in formal employment), whether the participant was currently in school, partnership status, and corresponding baseline values for each outcome examined (contraceptive use, health knowledge score, and social support score). All data analyses were conducted in R version 4.3.1.

**Qualitative analysis.** Post-intervention interview transcripts and WhatsApp chats (text, audio, media) were analyzed using a traditional content analysis approach [47]. The goal of qualitative analysis was to contextualize the way participants engaged and perceived their engagement with the FAMA intervention. Data were analyzed in Spanish to preserve their full meaning and context. Selected quotes were translated and reviewed for accuracy by bilingual study team members. Post-intervention interviews were analyzed using Dedoose, and WhatsApp chats were coded and analyzed in an Excel spreadsheet.

Post-intervention interviews had previously been coded and analyzed as part of evaluation of the FAMA intervention [41]. The aims of the present analysis were distinct from those of the initial report. As such, this analysis adapted the codebook used in the prior analysis to include companionship, informational, emotional, and instrumental social support [48], and focus on other key aspects related to engagement, rather than intervention feasibility and acceptability. A version of this new deductive codebook was adapted for use to quantify WhatsApp messages based on content and message type. The WhatsApp codebook included additional codes for identifying intervention-specific message content including emojis of the day and group forum responses. WhatsApp chat text and interviews were analyzed by a team of four bilingual (English and Spanish) coders. Subjective assessment [49] was conducted for both interviews and WhatsApp chats to ensure inter-coder agreement in codebook application; discrepancies were discussed after the coding team coded the same subset of transcripts and WhatsApp chats until resolution was reached. WhatsApp chats were separated by intervention group, and then organized into week-long sections of text to reduce coder burden. Each week-long section was distributed among coders and analyzed separately, but coders had access to the entire group chat to ensure the appropriate context was provided for each selection. Every sixth chat section was double coded to ensure consistency in coding. We employed several iterative data reduction techniques to identify relevant categories. First, the lead author read excerpts for each individual code and developed code summaries. Using these code summaries, the lead author then wrote analytic memos to identify patterns arising in the data.

**Integration of qualitative and quantitative data.** WhatsApp chats and post-intervention interviews were coded using complementary codebooks, allowing for codes to be applied across both types of data. To triangulate findings across the different sources of data, we compared and cross-validated insights from interviews and WhatsApp chat data. Using analytic memos, qualitative findings were integrated with quantitative messaging patterns to triangulate and explicate findings from both sets of data.

## Results

### Participant characteristics

A total of 58 participants enrolled in the FAMA intervention: all completed baseline questionnaires. Of these, 44 completed pre and post health knowledge questionnaires, we were able to obtain information about both pre and post contraceptive use for 53 participants, and 27 participated in post-intervention qualitative interviews. Table 1 summarizes participant baseline characteristics. Participants ranged from 15-20 years old at intervention baseline, and the median age of participants in the intervention was 18 years (interquartile range [IQR] 17–18 years). Although participant eligibility was limited to 14–19 year olds, some participants had turned 20 by the time the group intervention started and were still included. The median infant age was three months (IQR: 1–5 months) at the start of the intervention period, with a full age range of 0–11 months. Most participants had legal documentation (53, 91.4%) and over half had government-issued identification cards (36, 62.1%), which are both indications of experience navigating government systems. Few reported any employment (2, 3.5%) and

**Table 1.  Participant baseline characteristics.**

|  | Total (N) | n (%) or median (IQR) |
|---|---|---|
| Participant Age (Years) | 58 | 18.0 (17.0-18.0) |
| Infant Age (Months) | 57 | 3.0 (1.0-5.0) |
| Declared Status | 58 | 53 (91.4%) |
| Government-Issued ID | 58 | 36 (62.1%) |
| Attend School | 58 | 23 (39.7%) |
| Highest Grade Level | 58 | 9.0 (8.0-11.0) |
| Partnered | 58 | 46 (79.3%) |
| Partnership Duration (Years) | 46 | 2.0 (1.0-3.0) |
| Legally Married | 46 | 2 (4.4%) |
| Employed | 58 | 2 (3.5%) |
| Number of Children | 58 | 1.0 (1.0-1.0) |
| Contraceptive Use | 53 | 33 (62.3%) |
| Health Knowledge score | 44 | 7.0 (7.0-8.0) |
| Social Support score | 39 | 255.6 (239.1-266.7) |

23 (39.7%) were currently attending school. Most participants (46, 79.3%) reported currently having a romantic partner, and almost all were first time mothers (54, 93.1%).

## Intervention engagement

Median levels of each domain of engagement with the FAMA intervention are provided in Table 2. Participants had a median Adherence Index score of 0.4, indicating they acknowledged the moderator message and sent the emoji of the day a median of 40% of intervention days. The median Diligence Index score of 0.2 reflects participant responses to a median of 20% of forum questions, while the median Consistency Index score of 0.6 shows that participants sent one or more text messages a median of 60% of the intervention days. The median Support Index score of 0.5 suggests that a half of participant messages involved social support exchange, and the Focus Index score of 0.1 indicates that a median of 10% of participant messages were on-topic (either responding to daily message or centered on one of six FAMA topics). As the Immersion Index is calculated as a percentile, its median value is set at 0.5 and therefore not reported in Table 2. The number of messages sent by participants varied from 3 to 1026 messages over the intervention period. Overall, the median Engagement Index score for participants was 2.4 (IQR 1.8-3.1), out of a possible score of 6 points (which would have indicated perfect adherence, consistency, and diligence, the highest messaging in the group, and always sending on-topic messages containing a social support exchange).

## Participant-reported and observed messaging topics

Only 13.3% of total participant messages were coded as topically related to the most recent FAMA daily message, and 12% were related to any of the six FAMA topics (family planning, baby growth and development, breastfeeding, baby illnesses, feeding baby, and understanding baby). Messaging related to the six different content areas ranged from 1.5% (family planning) to 3.7% of total messages (baby illnesses) (Table 3).

During post-intervention interviews, participants were asked about the importance of different content areas and what they learned because of FAMA. Participants mentioned all six topics in interviews. Family planning was mentioned most frequently, by 21 of the 27 interviewees (78%), yet this topic made up the fewest number of messages exchanged (1.5% of

**Table 2. Engagement metrics.**

| Index | Median (IQR) | Mean (SD) |
|---|---|---|
| Adherence Index | 0.4 (0.1-0.7) | 0.4 (0.3) * |
| Diligence Index | 0.2 (0.2-0.3) | 0.3 (0.2) * |
| Consistency Index | 0.6 (0.3-0.8) | 0.5 (0.3) * |
| Support Index | 0.5 (0.4-0.6) | 0.5 (0.1) |
| Focus Index | 0.1 (0.1-0.2) | 0.2 (0.1) * |
| Engagement Index | 2.4 (1.8-3.1) | 2.4 (0.9) * |

*Variable is not normally distributed.

**Table 3. WhatsApp chat message content.**

| Statistic | n (%) |
|---|---|
| **N=17,808 total messages** | |
| Total Participant Messages | 14,093 (79.1%) |
| Message Topics | |
| Family Planning | 267 (1.5%) |
| Breastfeeding | 381 (2.1%) |
| Baby Illnesses | 658 (3.7%) |
| Growth and Development | 569 (3.2%) |
| Feeding Baby | 345 (1.9%) |
| Understanding Baby | 280 (1.6%) |
| Any of 6 Topics | 2,131 (12.0%) |
| On-topic* | 2,357 (13.3%) |
| Forum Response | 156 (0.9%) |
| Social Support Messages | |
| Companionship Support | 8,238 (46.3%) |
| Emotional Support | 294 (1.7%) |
| Informational Support | 1,393 (7.8%) |
| Instrumental Support | 83 (0.5%) |

*Same topic as most recent moderator message.

total messages across all three groups). Even the least mentioned FAMA topic—baby growth and development—was mentioned by 13 interviewees (48%). Most participants talked about multiple topic areas being particularly helpful for them:

> *[Original Spanish] Si porque aprendí, aprendí muchas cosas de lo que enviaban, cosas que no sabía, como por ejemplo, que yo tenía que a los 2 meses se le daba habichuelas [a mi bebé], cosas así y me decían que no, como amamantarla, que no tenía ninguna experiencia, me enseñaron, entre otras cosas.*

> *[English Translation] Yes because I learned, I learned many things from what they sent, things I didn't know, like for example that I had to, at 2 months I was feeding [my baby] beans, things like that and they told me no, how to breastfeed her, I didn't have any experience, they taught me, among other things. - Group 3 Participant, Age 18*

## Participant-reported and observed domains of social support exchanged

Of 17,808 messages exchanged, 46.3% were coded as exchanging companionship support, 7.8% as informational support, 1.7% as emotional support, and 0.5% as instrumental support (Table 3). The distribution was similar between the three groups. Table 4 provides example messages demonstrating each domain of social support as well as interview quotes highlighting the value of these domains.

**Table 4. Selected WhatsApp messages and interview quotes.**

| | | Quote ID | Quote (English Translation) |
|---|---|---|---|
| Companion-ship | WhatsApp Messages | 1.1 | *Mine is the same way, when it's nighttime she wakes up, one can't sleep, one has to stay awake watching her or else she will cry, mine is the same.* |
| | | 1.2 | *Oh [Name] but your boy is so handsome and very big, may God bless him, God blessing I want mine to be like that already.* |
| | Interviews | 1.3 | *Well, I felt more connected with them because I talked about things with them, it was fun talking with them, and they talked about good things.* |
| | | 1.4 | *Oh, I met a lot, a lot of friends, we talked about becoming friends. We are going to keep going out.* |
| | | 1.5 | *Honestly I liked the group, I liked it, I liked it, I liked it because [the moderators] made me really happy, really happy and honestly that has made me feel really bad that we won't be able to share anymore, but whatever I need I will have their numbers so whatever comes up I will write them.* |
| Informational | WhatsApp Messages | 2.1 | *Another question, if I have, I don't know, fever, body aches, and that type of thing, can I pass it to my baby through breastfeeding?* |
| | | 2.2 | Participant A: *Don't worry about vomiting, if baby vomits a lot they will get fatter.*<br>Participant B: *No my love, it does not make babies get fatter, because my daughter has been vomiting for almost a month, everything she eats she vomits, she doesn't eat, she doesn't want anything to eat, she is very skinny, it's not because she is going to get fatter, she has a cold and she got over her cold but the only thing she hasn't gotten over is the vomiting.* |
| | Interviews | 2.3 | *Mhm because my baby…before he didn't vomit, but now he is vomiting and I have asked [the moderator] and she told me that I had to come to the pediatrician to see what it is, and [the moderator] told me that that is somewhat normal or it's because of something that I am eating…* |
| | | 2.4 | *As a first-time mother, [FAMA] showed us how to attend to our babies, how to take care of them, how to bathe them, how to put their clothes on, how to feed them, how many times you have to give them food, and that, that was it.* |
| | | 2.5 | *[Fellow participants] helped me in things that they knew, they helped me, and I helped them.* |
| | | 2.6 | *This program is very good because it helps me learn many things, things that I thought were true, and are actually peoples' myths.* |
| | | 2.7 | *Yes, with my sisters and friends, and I shared [what I learned from FAMA] for the good because I have brought, I have come here with my friends who have come to plan, to get HIV tests and pregnancy tests and so on.* |
| Emotional | WhatsApp Messages | 3.1 | *Oh yes, I am shattered because now all my girl's things have burned, all of my things, everything, I don't have anything now, not even a place to sit, nothing. Everything was burned, I am shattered.* |
| | | 3.2 | *Well, personally, I get very stressed because when my baby sleeps, what I do is the chores, clean the house always, and when I want to sleep, he wakes up, and I get very stressed.* |
| | | 3.3 | *By tomorrow you will see that [your baby] will be better [from their illness]* |
| | Interviews | 3.4 | *I have had a lot of problems with my relationship with the father of my baby, and the moderators have helped me a lot.* |
| | | 3.5 | *...but even so, I felt like I was with family, I felt like I was in the warmth of a home with the girls.* |
| Instrumental | WhatsApp Messages | 4.1 | *All who are able, you can bring what you are going to give to [the participant] here at MAMI, and I, I will call you [participant] so you can come and pick up what I have for you in the meantime. But during this week, starting this afternoon until tomorrow I will get you more things, at least you'll have some pampers and some other clothes. -Moderator* |
| | | 4.2 | *[Moderator] I want you to make an appointment for me to take the boy to the doctor, please do that errand for me.* |
| | Interviews | 4.3 | *Well…it was something very beautiful. I participated a lot, they gave me a lot of support when my home burned down, they shared with me, they sent things, they're all good people. The moderators helped us too. I felt very supported, even my self-esteem increased. It was a very beautiful thing.* |

Of the four social support domains, companionship support messages were the most observed in the WhatsApp chats (46.3% of all messages were sharing companionship). WhatsApp messages revealed participants relating to each other's experiences, sharing personal stories, commiserating, and celebrating their babies (Table 4, Quote 1.1, 1.2). Interviewees talked about their enjoyment of "talking" with the group—sending and reading messages in the group, and even sending pictures of their babies each week (Table 4, Quote 1.3). In addition, some interviewees mentioned that they felt they had more friends and people to share enjoyable activities with because of FAMA, not only through WhatsApp, but also in-person. (Table 4, Quote 1.4) This suggests that the FAMA intervention played a key role for these participants in allowing them to connect with each other and overcome experiences of isolation. While interviewees spoke more frequently about companionship support from fellow participants, they also reported that moderators provided extensive companionship support by playing, joking, and making participants feel at home in the group (Table 4, Quote 1.5).

While analysis of WhatsApp chats showed only 7.8% of messages involved informational support exchanged by participants, this domain was mentioned by all but two interviewees. Participants raised informational questions (Table 4, Quote 2.1) and shared relevant experiences with each other (Table 4, Quote 2.2). In interviews, participants reported specific content areas they got help with (Table 4, Quote 2.3, 2.4) and discussed the exchange of information between them (Table 4, Quote 2.5). Interviewees reported the information often exposed their long-held beliefs as myths, and they were proud to be able to correct myths and provide informational support to their peers (both inside and outside of the study), partners, and even their own mothers because of what they learned in FAMA. (Table 4, Quote 2.6).

Emotional support was shared in 1.7% of messages, largely in the form of an empathetic exchange when participants were facing a difficult or emotional situation. From experiences with postpartum depression, to the loss of a family member, to the difficulties of being a new parent—participants turned to the group for emotional support (Table 4, Quote 3.1, 3.2). Fellow participants reassured each other in difficult situations (Table 4, Quote 3.3). In interviews, participants described emotional support as being provided by both moderators and fellow participants (Table 4, Quote 3.4 and 3.5).

Few messages communicated instrumental support: 0.5% of messages overall, of which over a third came from moderators (0.2% of overall messages) (Table 3). Examples of instrumental support exchange included requesting and receiving help in setting up medical appointments or discussing donations being made to a participant who had lost their home to a fire (Table 4, Quotes 4.1-4.2). Both moderators and participants were cited as being involved in the exchange of instrumental support, though these exchanges were rarely discussed in interviews (Table 4, Quote 4.3).

In comparing participant-reported and observed domains of social support, we generally observed consistency between participant messages and interviews, with slight discrepancies in the relative magnitude of support types. Interviewees spoke about informational support most frequently, followed by emotional support, companionship, and instrumental support. In contrast, in the chats, companionship messages were exchanged much more frequently than informational support, followed by emotional and instrumental support.

## Association between engagement in FAMA messaging and outcomes

We examined associations between messaging engagement and each FAMA outcome: health knowledge, contraceptive uptake, and social support (Tables 5–7).

**Table 5. Correlates of post-test health knowledge survey score.**

| | N | Unadjusted coefficient | 95% CI | Unadjusted p-value | Adjusted coefficient* | 95% CI | Adjusted p-value |
|---|---|---|---|---|---|---|---|
| Adherence Index | 44 | **3.17** | **1.05 – 5.28** | **0.01** | **2.84** | **0.83 – 4.84** | **0.01** |
| Diligence Index | 44 | 1.02 | -1.59 – 3.63 | 0.45 | 0.28 | -2.47 – 3.03 | 0.84 |
| Consistency Index | 44 | 2.28 | -0.25 – 4.81 | 0.09 | 1.46 | -1.10 – 4.02 | 0.27 |
| Support Index | 44 | -3.39 | -8.29 – 1.51 | 0.18 | **-5.78** | **-10.42 - -1.00** | **0.02** |
| Immersion Index | 44 | 1.54 | -0.50 – 3.58 | 0.15 | 1.09 | -0.92 – 3.10 | 0.29 |
| Focus Index | 44 | 4.42 | -3.00 – 11.83 | 0.25 | 4.45 | -2.71 -11.61 | 0.23 |
| Engagement Index | 44 | **0.76** | **0.05 – 1.47** | **0.04** | 0.53 | -0.19 – 1.25 | 0.16 |

*Adjusted for baseline health knowledge score, participant age, declared status, school status, partnership status

Forty-four participants completed both the pre- and post-intervention health knowledge survey. The median change in health knowledge from enrollment to follow-up was +1.00 point (IQR: 0.00-2.25 points) out of a maximum of 11 points [41]. We found a significant association between Engagement Index and health knowledge score in unadjusted analyses: each 1-unit increase in the Engagement Index (out of a maximum score of 6) was associated with a 0.76-point higher health knowledge score (out of a maximum score of 11; CI: 0.05-1.47, p = 0.04). This association did not remain statistically significant after adjustment for confounding variables. In analyzing the individual indices that make up the composite Engagement Index, we found that the Support Index and the Adherence Index were individually associated with health knowledge scores. In multivariate analysis adjusted for confounders, a 20% increase in Adherence Index score was associated with a 0.57-point higher health knowledge score (p = 0.01). In contrast, a 20% increase in Support Index was associated with a 1.16-point lower health knowledge score (p = 0.02) (Table 6).

Table 6 summarizes analysis of the association between engagement and contraceptive use. At follow-up, 53 of 58 participants had medical record data on contraceptive use, of whom 41 (77.4%) were using some form of contraceptive [41]. We found no significant association between Engagement Index or any of the component indices with contraceptive use at exit. Table 7 summarizes analysis of the association between engagement and social support score. Thirty-nine participants had social support survey data available at follow-up. We found no significant association between the Engagement Index and post-intervention social support scores.

**Table 6. Correlates of contraception use at end of intervention.**

| | N | Unadjusted RR (95% CI) | Unadjusted p-value | Adjusted RR* | Adjusted p-value |
|---|---|---|---|---|---|
| Adherence Index | 53 | 1.03 (0.68-1.58) | 0.87 | 1.03 (0.73-1.44) | 0.87 |
| Diligence Index | 53 | 1.29 (0.85-1.95) | 0.24 | 1.35 (0.79-2.32) | 0.27 |
| Consistency Index | 53 | 1.25 (0.77-2.04) | 0.37 | 1.22 (0.80-1.86) | 0.37 |
| Support Index | 53 | 0.57 (0.19-1.67) | 0.30 | 1.31 (0.56-3.04) | 0.53 |
| Immersion Index | 53 | 1.06 (0.67-1.67) | 0.82 | 1.12 (0.77-1.65) | 0.55 |
| Focus Index | 53 | 0.52 (0.09-3.14) | 0.48 | 0.32 (0.07-1.53) | 0.15 |
| Engagement Index | 53 | 1.03 (0.90-1.18) | 0.67 | 1.05 (0.92-1.19) | 0.48 |

*Adjusted for baseline contraception use, participant age, declared status, school status, partnership status.

**Table 7. Correlates of social support survey score.**

| | N | Unadjusted coefficient | 95% CI | Unadjusted p-value | Adjusted coefficient* | 95% CI | Adjusted p-value |
|---|---|---|---|---|---|---|---|
| Adherence Index | 39 | -17.10 | -38.10 – 3.89 | 0.12 | -17.45 | -37.67- 2.77 | 0.10 |
| Diligence Index | 39 | 0.27 | -24.11 - 24.66 | 0.98 | 2.08 | -22.00- 26.17 | 0.87 |
| Consistency Index | 39 | -10.17 | -33.86 - 13.53 | 0.41 | -11.65 | -33.88- 10.57 | 0.31 |
| Support Index | 39 | -9.52 | -53.93 – 38.89 | 0.68 | 1.05 | -43.54 – 45.63 | 0.96 |
| Immersion Index | 39 | -4.79 | -23.22 – 13.63 | 0.61 | -8.00 | -24.68 - 9.68 | 0.38 |
| Focus Index | | -30.55 | -97.26 – 36.16 | 0.38 | -33.56 | -97.83 – 30.71 | 0.31 |
| Engagement Index | 39 | -3.62 | -10.28 – 3.04 | 0.29 | -3.73 | -10.00 – 2.55 | 0.25 |

*Adjusted for participant age, declared status, school status, partnership status, and baseline social support score.

## Discussion

In this study, we found that adolescent mothers largely used the FAMA moderated WhatsApp peer group to share companionship with other adolescent mothers, as well as get informational support related to their and their babies' health. Emotional and instrumental support were exchanged less frequently. In qualitative interviews, participants highlighted the importance of informational and emotional support more often than was reflective of the observed frequency of these messages.

Although most participant messages were not explicitly tied to the six key content areas discussed in FAMA intervention messages, participants reported gaining important knowledge and learning new skills related to caring for their infants. The most common FAMA topics discussed in the groups were baby illness, growth and development, and breastfeeding. Family planning was the topic mentioned most during interviews but least discussed in the WhatsApp chats themselves. This could indicate that although this was an important topic, participants may have felt less comfortable talking about this in the group.

To probe the potential mechanism of action of the FAMA intervention, we evaluated the association between engagement and health knowledge, social support, and contraceptive use, hypothesizing that greater engagement would be associated with improved outcomes. Based on our hypothesized mechanism (Fig 1), we hypothesized that engagement with the daily informational messaging aspects of FAMA (measured by adherence, diligence, and focus) would be associated with greater changes in health knowledge, while engagement with the peer group chat (measured by consistency, immersion, and support) would be associated with greater changes in social support scores. We additionally hypothesized that the overall Engagement Index would be associated with changes in health knowledge, social support score, and uptake of contraceptives. We found that greater intervention engagement, as measured by acknowledging moderator messages with the "emoji of the day" (adherence) was associated with higher health knowledge scores at follow-up. In contrast, engagement as measured by a greater proportion of messages involving exchange of social support, was associated with lower health knowledge. We found no association between engagement and contraceptive use or social support.

Our findings however, do reinforce the significance of digital companionship for young mothers [50]. This is consistent with pre-post analysis of the FAMA intervention that showed an increase in contraceptive use, clinic visit attendance, and evidence of improved social connection. The main role of peers in this group was in exchanging companionship support with one another, and participants frequently emphasized its value in their interviews. Participant interviews suggested that overcoming the isolation of adolescent motherhood by connecting with peers was a vital part of the appeal of this intervention to participants. The predominance

of companionship support in the WhatsApp chats suggests that young mothers may have prioritized peer companionship connections over other forms of support—such as informational or instrumental support. This may in part be due to the unique social and emotional context of adolescent motherhood. Given that isolation is a common experience of adolescent mothers, [50,51] it is possible that opportunities to share experiences and foster a sense of community were more immediately relevant and accessible for this group. However, our analysis found that while companionship was a key engagement style, the exchange of social support more broadly was not found to be positively associated with changes in intervention outcomes. Importantly, social support exchange was found to be negatively associated with health knowledge at posttest. This suggests that while companionship may fulfill an important social need, it does not necessarily facilitate changes in behavior and may even pose a challenge for observing changes in health knowledge.

In previously published work, we found that the FAMA intervention was associated with increased health knowledge and contraceptive use [41]. Although we did not find that engagement with the intervention was associated with contraceptive use, our finding of significant association between a measure of engagement and health knowledge score aligns with prior research and suggests a potential dose-response relationship between intervention engagement and increased knowledge. However, given the study's sample size and power limitations, these findings should be interpreted with caution and warrant further investigation in larger studies. Among the different types of engagement, we found that more "adherence"— sending the required emoji of the day—was positively associated with changes in health knowledge. Interestingly, this association did not apply to the Consistency Index, which measured the proportion of days participants sent a text message. This may reflect that sending the accurate emoji of the day involves a closer reading of the daily informational message, whereas sending a message each day could have been achieved by sending a greeting in the morning without engaging with any of the informational material from the intervention.

Our analysis also revealed an unexpected finding: the exchange of social support was negatively associated with posttest health knowledge. This may be explained by participants using the FAMA intervention primarily as a space to chat and engage with peers, which could have distracted from the educational messages shared. This social dynamic may have shifted focus away from educational materials and towards peer relationships themselves. Another potential explanation could be that adolescents who understood intervention content more fully had less need to interact in the group. This finding warrants further study to better understand the relationship between social support exchange and knowledge acquisition in digital health interventions.

In contrast to the associations we found with health knowledge, we did not detect an association between engagement and contraceptive use or social support. Other mHealth studies have similarly found that higher user engagement may not always translate to changes in behavior [52–54], especially as behavior change has more complex drivers than knowledge [55,56]. Additionally, family planning comprised only one sixth of intervention-scripted messages and was the last topic to be discussed, so it is perhaps unsurprising that engagement over all topics and throughout the intervention would not be associated with contraceptive uptake. The lack of association with social support scores is consistent with the finding of the initial FAMA study that none of the social support sub-scales changed significantly from pre- to post-test for study participants [41].

A number of mHealth interventions have been developed for maternal child health, some using SMS to deliver informational messages [57,58] or mediate one to one connection between mothers and healthcare workers [59,60] or peers [26]; others using an intervention-specific phone application to deliver information [25,46,53]; and still others

incorporating an in-person group component [27] or a group chat feature to allow for peer interaction [24,34,61–63]. While most interventions are aimed at adult mothers or first-time mothers broadly, our study adds to the growing literature focused specifically on adolescent mothers around the world [24,26,27,64–66]. Our analysis is consistent with previous findings that mHealth interventions are well-used and well-received by adolescent mothers in the United States, Canada, and Zambia [24,64,66]. Similar studies have found that social support in general does play a key role in these interventions [64,66]. Our analysis extends these findings to suggest that a close reading of message content may have a greater effect on knowledge, as measured by pre and posttest, than other forms of engagement.

User engagement in mHealth interventions is complex, which makes it difficult to quantify [67]. However, there is a need to better understand and optimize user engagement to improve the success and utility of these interventions [67]. Recognizing this need, some maternal child health interventions have begun to measure and explore this construct [34,46,53]. A recently published study examined user engagement with Milk Man—a mobile app designed for engaging new parents with information on breastfeeding and parenting in Western Australia [53]. These authors incorporated engagement metrics such as frequency of app use, number of library articles read in the app, and user feedback on satisfaction. To our knowledge, this is the only other maternal and child health intervention to examine the association between user engagement and health outcomes. Similar to our study, the authors found that engagement with the Milk Man app was not associated with their behavior change of interest, exclusive breastfeeding [53]. Another recent study of a WhatsApp group for mothers of all ages in Southern Brazil [34] mined messages sent by mothers, calculating topic and message frequencies for times of day and days of the week to determine the most active moments in their WhatsApp-based intervention groups. Similar to our analysis, these studies allowed for a variety of user engagement types, recognizing the inherent complexities of engaging with mobile health interventions [46,67,68]. Our study builds on previous work by developing additional methods for analyzing user engagement on WhatsApp or other group messaging platforms which are limited in their engagement measurement capabilities. In addition, our study examined the association of different facets of engagement and multiple study outcomes including not only behavior but knowledge and perceived social support. This demonstrates an important next step in leveraging the study of user engagement for intervention enhancement as it allowed us to identify the specific types of engagement that were most strongly linked to more proximal outcomes such as knowledge and perceived social support. Our finding of opposing associations with outcomes using different engagement measures additionally highlights the importance of considering multiple measures of engagement.

A strength of this study was that it was conducted using mixed methods and comparing observed messaging and reported experiences to give a more nuanced understanding of participant engagement. The minor discrepancies observed when comparing these two data sources underscores the strength of using mixed methods as we were able to identify potential gaps between observed engagement and reported needs – insights that can help design more responsive digital health programs tailored to participant needs. We also developed novel measures of engagement in a WhatsApp group, considering different dimensions of engagement. All members of the coding team were bilingual Spanish and English speakers, allowing us to preserve as much of their original meaning as possible. In addition, members of the implementation team were involved in this analysis as senior advisors and coders, to ensure the subtleties of the research context were understood and captured.

Our study also has limitations. Our sample size was relatively small (58 participants), and the intervention was only conducted in one clinic in the Dominican Republic. Our findings may not be applicable to other regions or more diverse populations. Future research should

aim to replicate these findings in larger, multi-site studies to improve generalizability and assess the FAMA intervention in different settings. Additionally, participants were recruited from a maternal health clinic, which could introduce selection bias into our study. Adolescents who visit this clinic may be more likely to be more engaged with health services than other adolescent mothers in the region, potentially skewing our sample towards individuals with greater baseline health literacy and a higher likelihood of engaging in the intervention. Future studies could address this issue by recruiting from multiple types of community settings to include a broader range of participants. We also faced limitations in the quality of data gathered in this study. We had high levels of missing outcome data, which may have biased our analyses as those whose information was incomplete may have been those least engaged with the intervention. This could have resulted in an overestimation of the association between engagement with FAMA and identified outcomes. This issue may be addressed in future studies through the implementation of more robust follow-up strategies. Multimedia messages were not included in our analysis, which may have resulted in under-reporting of exchanges of social support and FAMA topic-related messaging. Future studies should prioritize including all forms of communication to provide a more comprehensive picture of engagement. While we attempted to capture multiple dimensions of engagement in our analysis, our measures have not been previously validated and may not capture all forms of engagement. For example, we have no data on participants reading messages without sending a response, which may be a significant form of engagement that was left unmeasured. In addition, we cannot distinguish between those who truly read the message and those who simply skimmed and sent the corresponding emoji of the day. This could have resulted in misclassification of participants engagement, however, as we were unable to ask the participants about this directly, it is difficult assess the degree and direction of this potential bias. As there are no existing mechanisms to measure this kind of engagement in WhatsApp, future studies may explore new ways to approximate this type of user engagement. Another important limitation is that in all three of the WhatsApp chat groups, some members decided to make another, separate chat group just for participant messaging, with no rules and without the participation of the moderators. The existence of these outside groups limits our ability to assess how social support was exchanged and how participants engaged within these groups as we only had access to the moderated intervention chat groups. Future iterations of this or similar work may consider ways to prevent the creation of additional outside groups until after study completion.

In conclusion, this analysis supports the hypothesis that engagement with FAMA digital informational content was associated with increases in knowledge. It also adds to our understanding of the ways users engage with group mHealth interventions and contributes new approaches to measure engagement, accounting for the different engagement styles participants may have. Future interventions may leverage our findings about the importance of close reading of informational messages when making decisions about intervention development for this population, for example by designing interventions to promote this type of engagement. Interventionists should also consider designing programs that balance opportunities for peer connection with strategies that encourage deeper engagement with informational content. For example, creating designated time for educational content within or even separate from peer interactions could help to foster a better balance between social connection and health knowledge acquisition.

While our study provides valuable insight into engagement patterns, continued study is needed to further probe the potential mechanisms of digital health interventions and assess their long-term impact and risks. Longitudinal studies may be particularly valuable in examining the sustained effects of digital interventions on maternal health outcomes over time. These

kinds of studies could also help determine whether initial gains in knowledge translate into lasting behavioral changes and examine how engagement patterns evolve over time. This line of research could inform the development of even more effective strategies for improving maternal health using digital platforms.

## Supporting information

**S1 Data.  Study data.** Deidentified study questionnaire data are included. WhatsApp data are not included as they could not be deidentified.
(CSV)

## Acknowledgements

We would like to thank Vivian Araujo, Leidy Soriano, Kari Aquino, Diane Bushley, and Eliza Davison for their work on the FAMA intervention development, implementation, and analysis.

## Author contributions

**Conceptualization:** Alana Roberta Lopez, Elizabeth Haight, Jennifer Unger, Mina Halpern, Keshet Ronen.

**Formal analysis:** Alana Roberta Lopez, Elizabeth Haight, Linda Guijosa, Briana Williams, Keshet Ronen.

**Funding acquisition:** Elizabeth Haight, Mina Halpern, Samantha Stonbraker.

**Investigation:** Elizabeth Haight, Linda Guijosa, Luz Messina.

**Methodology:** Alana Roberta Lopez, Elizabeth Haight, Jennifer Unger, Mina Halpern, Keshet Ronen.

**Project administration:** Elizabeth Haight, Linda Guijosa, Luz Messina, Mina Halpern, Samantha Stonbraker.

**Supervision:** Elizabeth Haight, Jennifer Unger, Mina Halpern, Samantha Stonbraker, Keshet Ronen.

**Validation:** Alana Roberta Lopez, Luz Messina, Mina Halpern, Keshet Ronen.

**Visualization:** Alana Roberta Lopez.

**Writing – original draft:** Alana Roberta Lopez, Keshet Ronen.

**Writing – review & editing:** Alana Roberta Lopez, Elizabeth Haight, Linda Guijosa, Briana Williams, Jennifer Unger, Luz Messina, Mina Halpern, Samantha Stonbraker, Keshet Ronen.

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
