## [Decision Letter · Decision Letter 0]

16 Dec 2024

PDIG-D-24-00488Experiences of social support and the role of engagement in a digital educational support group for adolescent mothers’ health in the Dominican RepublicPLOS Digital Health Dear Dr. Lopez, Thank you for submitting your manuscript to PLOS Digital Health. After careful consideration, we feel that it has merit but does not fully meet PLOS Digital Health's publication criteria as it currently stands. Therefore, we invite you to submit a revised version of the manuscript that addresses the points raised during the review process. Please submit your revised manuscript within 60 days Feb 14 2025 11:59PM. If you will need more time than this to complete your revisions, please reply to this message or contact the journal office at digitalhealth@plos.org. Please include the following items when submitting your revised manuscript:* A rebuttal letter that responds to each point raised by the editor and reviewer(s). You should upload this letter as a separate file labeled 'Response to Reviewers '. This file does not need to include responses to any formatting updates and technical items listed in the 'Journal Requirements' section below.* A marked-up copy of your manuscript that highlights changes made to the original version. You should upload this as a separate file labeled 'Revised Manuscript with Track Changes '.* An unmarked version of your revised paper without tracked changes. You should upload this as a separate file labeled 'Manuscript '. If you would like to make changes to your financial disclosure, competing interests statement, or data availability statement, please make these updates within the submission form at the time of resubmission. Guidelines for resubmitting your figure files are available below the reviewer comments at the end of this letter. We look forward to receiving your revised manuscript. Kind regards, Haleh AyatollahiSection EditorPLOS Digital Health Haleh AyatollahiSection EditorPLOS Digital Health Leo Anthony CeliEditor-in-ChiefPLOS Digital Healthorcid.org/0000-0001-6712-6626 **Additional Editor Comments (if provided):****Reviewers' Comments:** Reviewer's Responses to Questions

**Comments to the Author**

1. Does this manuscript meet PLOS Digital Health’s publication criteria ? Is the manuscript technically sound, and do the data support the conclusions? The manuscript must describe methodologically and ethically rigorous research with conclusions that are appropriately drawn based on the data presented.

Reviewer #1: Yes

Reviewer #2: Yes

Reviewer #3: Partly

Reviewer #4: No

Reviewer #5: Yes

2. Has the statistical analysis been performed appropriately and rigorously?

Reviewer #1: Yes

Reviewer #2: Yes

Reviewer #3: Yes

Reviewer #4: Yes

Reviewer #5: No

3. Have the authors made all data underlying the findings in their manuscript fully available (please refer to the Data Availability Statement at the start of the manuscript PDF file)?

Reviewer #1: Yes

Reviewer #2: Yes

Reviewer #3: Yes

Reviewer #4: No

Reviewer #5: Yes

4. Is the manuscript presented in an intelligible fashion and written in standard English?

Reviewer #1: Yes

Reviewer #2: Yes

Reviewer #3: Yes

Reviewer #4: No

Reviewer #5: Yes

5. Review Comments to the Author

Reviewer #1: This is a well written manuscript presenting mixed-methods secondary analysis data. The research question is difficult to answer in a quantifiable way, this has been addressed in the discussion, and I agree a broad approach is needed.

Reviewer #2: This is an excellent manuscript on design, implementation and rigorous evaluation of a an intervention for education and support of adolescent mothers in Dominican Republic. The intervention was targeted to educate and support mothers at high risk of depression, lack of future contraception and not being able to properly care for their babies due to lack of knowledge. The analysis is very detailed and rigorous. The authors realize they have a small sample size but they were able to draw valid conclusions that can help with future implementation of similar support systems.

The attention to detail and rigorous both quantitative and qualitative analysis of the study are outstanding.

The study demonstrates the importance of providing emotional and educational support to high risk adolescent mothers.

Thank you for your submission.

Reviewer #3: The authors present the results of an intervention to improve sexual and reproductive health in a group of adolescent mothers in Dominican Republic. I personally liked the article. It touches an important issue that disproportionally impacts those settings that have the least access to appropriate healthcare. Thus, it is an important study to inform future approaches that aim to improve health access and information across these settings. However, the study could be improved in certain areas that I will describe below:

The problematic described in the abstract and introduction refers to the burden of giving birth at an early age. Thus, it may seem that the goal of the study is to improve contraception in adolescent girls to reduce the rates of early motherhood. Nevertheless, all the participants are already mothers. Maybe the described problematic could further emphasize why it is important to increase the use of contraception in mothers, are the number of children significantly higher in those women that had their firstborn during adolescence? Are the markers of risk of depression and low educational attainment significantly associated with number of children? If yes, then it is important to say so. Otherwise, it is unclear why the efforts to improve contraception were focused on girls that were already mothers.

Another confusing point regarding the usefulness of the system, and its possible impact, involves contraception accessibility. The authors say that 27% of adolescents in the included age group have to overcome numerous barriers to access contraceptives. How does their system approach this? Is it possible that the use of contraception did not improve because the mothers could not access them anyway? This is something that requires further clarification. If they cannot access contraception, then this might not be a good outcome metric for the study.

Regarding the association results, it is puzzling that support index and knowledge were negatively associated. Is it possible that the girls that understood best the information would interact the least because they didn't need it? The opposite could also be true. A correlation analysis between the multiple indices could be useful to better understand the results of the study.

Finally, the authors need to improve the critical comparison of their approach to other similar approaches. Most of the end of the discussion is limited to listing other similar approaches without a quantitative comparison. It is important that the authors tell us how they are better or not than other approaches, by how much, and why. This would strengthen the impact of the paper beyond the negative results regarding contraceptive use.

Overall, I think this is a good study that does not require the use of fancy machine learning algorithms to convey its importance in the development of mHealth approaches. I commend the authors for their efforts.

P.S.: Some minor comments:

1. Table 2 in the results is highly redundant with the corresponding text. Maybe it would be better to limit the text to describe the most important aspect of the table, while the table provides all the numerical results on its own.

2. Also regarding table 2, it does not make sense to include the immersion index as and empty row. The authors could ignore it in the table and explain in the text that the average of this metric has no meaning.

Reviewer #4: I would like to extend my thanks for submitting your article for review. After a thorough evaluation, it is evident that the manuscript has some foundational challenges that need to be addressed to elevate its overall quality. While the paper presents some interesting points, it currently lacks the depth, clarity, and innovation expected for publication. However, I have provided specific recommendations and suggestions for improvement, which I believe, if implemented, will significantly enhance the manuscript's contribution to the field.

Abstract:

1. The objective of this section should be more explicitly stated. It is important to clearly outline the specific research question or hypothesis that this study addresses. For instance, the statement "We conducted a mixed-methods secondary data analysis" could be improved by emphasizing what the primary aim of the research was (e.g., "This study aimed to explore the mechanisms of action of the FAMA intervention through a mixed-methods secondary data analysis").

2. Provide more clarity about the methodology used. For example, the description of "linear or Poisson regression" should be expanded to explain why these methods were chosen and how they were applied in this specific context. A brief mention of how the regression models account for potential confounding factors would also enhance the clarity of the methodology.

3. The results presented (e.g., “a significant positive association between engagement as measured by acknowledging intervention messages...”) need to be contextualized more clearly. It would be helpful to explain what these results mean in practical terms and how they relate to the intervention's overall success. Specifically, interpreting the negative association with social support in the context of health knowledge is crucial.

Introduction:

1. The introduction should include a more robust theoretical framework that underpins the study. While it does an excellent job of presenting background information, integrating relevant theories (e.g., theories of engagement, social support, or mHealth interventions) would strengthen the rationale behind the study..

2. identification of the research gap is somewhat implicit. A more direct statement of the gap in existing literature would make the introduction stronger..

3. There are some areas where the transition between paragraphs could be smoother. Consider reorganizing the introduction to flow more logically from the global context, to the specific issues in the Dominican Republic, then to the importance of the intervention, and finally to the hypothesis.

Materials and Methods:

1. The description of the methods is detailed but lacks justification for some choices, such as the use of the "Engagement Index." It would be beneficial to elaborate on why this particular method was chosen to measure engagement and how it accurately reflects participant involvement in the intervention..

2. sample description should be expanded. While eligibility criteria are provided, more context about the recruitment process and participant characteristics would help readers better understand the scope and limitations of the findings..

3. Clarify the process of data collection, particularly the integration of WhatsApp chat data with interview data. It would be useful to explain how the qualitative and quantitative data were merged, particularly the specific steps taken during the analysis to ensure that both types of data informed the interpretation of the results.

4. Ethical approval and consent procedures are adequately described, but this section would benefit from further details about the confidentiality measures in place. For instance, explaining how participant anonymity was ensured when handling WhatsApp messages or interview data would strengthen the ethical transparency of the study.

5. While statistical methods are mentioned, there is no explanation of the rationale behind choosing linear regression versus Poisson regression for different outcomes. It would be beneficial to clarify the assumptions of each model and why they were the most appropriate for the data being analyzed.

Results:

1. The section lacks clarity in reporting participant characteristics. Ensure that the baseline characteristics, such as the age range, employment status, and level of education, are precisely defined. The age of participants is mentioned as a median with interquartile range, but it would be beneficial to include the full range (minimum and maximum) to provide a clearer picture of the demographic diversity..

2. The engagement metrics are reported with median values and interquartile ranges but could benefit from additional statistical tests to show variability and significance. Consider using means and standard deviations where appropriate, especially when the data are approximately normally distributed. Furthermore, include p-values or confidence intervals for key comparisons to strengthen the statistical reporting.

3. Some message content (e.g., WhatsApp messages) is included in the Results section, but they do not appear to add value or meaningfully contribute to the data analysis. The inclusion of direct message excerpts may distract from the key findings and potentially overwhelm the reader. Consider summarizing the types of messages exchanged rather than providing lengthy quotes, unless they are directly tied to specific analysis.

Discussion:

1. The discussion regarding the types of social support exchanged (e.g., companionship, informational, emotional, and instrumental) could benefit from a deeper analysis. Specifically, provide a more nuanced interpretation of why companionship support predominated in the WhatsApp chats, considering the context of adolescent motherhood. Does this suggest that companionship was more vital than other forms of support due to the isolation or emotional challenges faced by young mothers?.

2. The findings regarding the association between engagement and health knowledge are important but should be explored in greater depth. You mention that “greater engagement, as measured by acknowledging moderator messages with the 'emoji of the day,' was associated with higher health knowledge scores at follow-up.” Elaborate on why this form of engagement (emojis) was positively associated with health knowledge. Consider discussing potential psychological or motivational mechanisms that may explain why this form of engagement, which appears less substantive, was more meaningful for the participants than more consistent messaging.

3. The finding of a negative association between the Support Index and health knowledge warrants further exploration. Provide a detailed discussion of possible explanations for this unexpected result. Could it be that participants who engaged more in social support exchanges were less focused on the informational content? Was there a distraction effect caused by the peer-to-peer interaction, and if so, how might future interventions better balance companionship with educational content?

4. e study identifies "adherence" as a form of engagement, yet it’s not entirely clear why this particular measure was significant in predicting health knowledge scores. The discussion should clarify why this specific type of engagement (adhering to the moderator's daily prompts) was associated with improved health knowledge, and how this can inform future interventions. Do participants who show high adherence to prompts tend to engage more meaningfully with the content, or is this association merely a byproduct of the structure of the intervention?

5. The study's results contrast with previous findings, where FAMA was associated with increased health knowledge and contraceptive use. In the discussion, acknowledge this discrepancy and analyze potential reasons for this divergence. Were there external factors at play that could have impacted the outcomes (e.g., cultural differences, sample characteristics)? Provide a more thorough comparison with existing literature to situate your findings within the broader context of digital health interventions.

6. The limitations section is somewhat brief and should be expanded. For example, the small sample size and the single-clinic design are mentioned, but how might these limitations specifically impact the generalizability of the results? Were there any biases in participant selection or in the completion of outcome measures (e.g., who dropped out or who missed surveys)? Discuss how these limitations could affect the interpretation of the findings and suggest avenues for future research to address these gaps.

7. The discussion could be strengthened by including more specific recommendations for future studies. Based on the results, what additional data or approaches would be valuable to explore? Consider suggesting longitudinal studies to better assess the long-term impact of digital interventions on maternal health outcomes, or exploring alternative ways of measuring engagement that go beyond text-based communication (e.g., multimedia interactions, passive reading of content).

Reviewer #5: Dear Authors,

I would like to say first and foremost I enjoyed reading your article and the type of research you are doing. This research is so crucial for adolescent mothers and with the increased use of digital health to support knowledge acquisition and increased connections/communications with patients, this understanding of what needs they need help with is timely and pertinent. To this I agree findings from your manuscript would assist other clinics globally with helping this target population (generally but there would naturally be cultural variations).

To this, I wish to help your manuscript, but I am struggling with its arguments. In addition, the findings from your already published article are very similar to what you are writing here. There is something here, but the argument and points drawn need to be reshaped.

One of the struggles I believe you experienced was low power due to the low N of subjects enrolled and then the reduced numbers of outcome data received. This most likely impacted the effect size with finding the strength of relationships between variables or how large that difference was. As the power appears low, which is suggested by the low levels of statistical significance, I would be very careful to state your findings are conclusive but rather important findings that warrant further attention and research analysis. This is particularly true for Lines 425-427 “Our finding of significant association between a measure of engagement and health knowledge score bolsters the prior findings by supporting a dose-response relationship between intervention engagement and increased knowledge”. Because of the low N, it gets very tricky to state a conclusive exposure-response relationship as this would require a larger N or power from the study.

1. Knowledge acquisition may improve skills but changing behaviors is more complicated. Thus, to change a behavior, finding individual’s core needs and motivations for change will support increased engagement and ultimately attitude change.

2. Line 83- “Social support is key for adolescent mothers whose developmental stage is characterized by increased reliance on and trust of peers” Can you explain by the publications referenced how social support is key? Can you back it up possibly with a published quantitative value?s

3. Lines 84-86- “Adequate social support, particularly peer support has been associated with reduced perinatal depression, improved knowledge about child development, improved breastfeeding indicators” Can you provide the published numbers of reduction or improvement?

4. From Lines 382-383- Findings Support Index and the Adherence Index were individually associated with health knowledge scores. This warrants further understanding from your population – what were the factors within the social support that influenced their adherence or engagement and eventually supported knowledge acquisition. What changed their attitudes? Was anything found from the interviews about this? If not, can you go back and ask?

5. Lines 393-394 “We found no significant association between the Engagement Index and post-intervention social support scores”. I think the types of engagement got a bit mixed together since the focus seemed more on knowledge acquisition.

6. Lines 399-400 – “In qualitative interviews, participants highlighted the importance of informational and emotional support more often than was reflective of the observed frequency of these messages” This warrants further investigation and an important point as this could suggest what you are designing maybe missing their actual needs. A more detailed needs assessment could be conducted from this preliminary study. It shows that how you perceive success with the app may be different from how the target population perceives it as helpful and what they actually need. For example, in the Author summary you actually mentioned this-“future digital group interventions should consider diverse engagement styles and the specific needs of adolescent mothers.” I was hoping to get a better understanding of this in the manuscript.

7. Lines 419-421 “We found that greater intervention engagement, as measured by acknowledging moderator messages with the “emoji of the day” (adherence) was associated with higher health knowledge scores at follow-up.” I think it would hard to conclude if A then B. The higher health knowledge scores were noted but could you expand more on how you tested their knowledge to truly know if knowledge was attained? Some could have just “skimmed” they article to get to the emoji and then send the emoji in the message. Also, as you mentioned you do not know if they read the articles and then forgot to send the emoji. Did you ever ask them? Were you able to get a rough number per participant of read but non-emoji responses?

8. This is the same point here in Lines 458-459, “Our analysis extends these findings to suggest that a close reading of message content has a greater effect on knowledge than other forms of engagement.”

iii. Lines 466-467- “This study incorporated engagement metrics such as frequency of app use, number of library articles read in the app, and user feedback on satisfaction”. This could be explained in the methods section and design of the app in the introduction to help explain the knowledge the participants received.

9. You have tapped into something here with helping this target population who desire a community or a place where they feel they belong and are accepted especially since they are at risk for subjugation to stigmas and discrimination. Correlating to feeling accepted or belonging to a group (adolescent behavior of love and connection- need for social connection), can you reference further any data you may have collected on depression/postpartum depression, social isolation, and loneliness? Can you expand on these points? Did you collect any data on increased mother-baby bonding?

10. Line 408-409 –“Overcoming the isolation of adolescent motherhood by connecting with peers was a vital part of the appeal of this intervention to participants”- this needs to be expanded further in the article to provide the backup for its significance.

11. Lines 495-496 “some members decided to make another, separate chat group just for participant messaging, with no rules and without the participation of the moderators”. This I can imagine was frustrating as this was lost data. Going forward, it would be helpful to state in the informed consent that groups can be developed after the completion of the intervention but not during as the point is to help them. But their feedback is welcome at any point- could be done as an anonymous message or survey form for quality improvement.

12. Lines 507-508 “Interventionists should also consider the need for companionship among young mothers yet recognize its potentially neutral (or even negative) role in relation to changes in health knowledge.” This needs to be expanded further so explain how this conclusion was drawn.

13. This really is an important point and how our social cognition is changing to two different versions- one from having “in person” contact and the one from digital contact and online social exchanges. How will you be able to detect if a mother is experiencing post-partum depression and then possibly has negative interactions with a participant online. How would you be able to detect an issue or protect them from this? There is a great article I would recommend reading - Social cognitive neuroscience in the digital age - PubMed ( DOI: 10.3389/fnhum.2023.1168788)

I hope you find this helpful as you are revising your manuscript. I look forward to reading it again.

6. PLOS authors have the option to publish the peer review history of their article (what does this mean? ). If published, this will include your full peer review and any attached files.

**Do you want your identity to be public for this peer review?** For information about this choice, including consent withdrawal, please see our Privacy Policy .

Reviewer #1: No

Reviewer #2: **Yes: ** Laritza M. Rodriguez

Reviewer #3: No

Reviewer #4: No

Reviewer #5: No

---

## [Decision Letter · Decision Letter 1]

28 Feb 2025

Experiences of social support and the role of engagement in a digital educational support group for adolescent mothers’ health in the Dominican Republic

PDIG-D-24-00488R1

Dear Ms. Lopez,

We are pleased to inform you that your manuscript 'Experiences of social support and the role of engagement in a digital educational support group for adolescent mothers’ health in the Dominican Republic' has been provisionally accepted for publication in PLOS Digital Health.

Best regards,

Haleh Ayatollahi

Section Editor

PLOS Digital Health

**Additional Editor Comments (if provided):**

**Reviewer Comments (if any, and for reference):**

Reviewer's Responses to Questions

**Comments to the Author**

1. If the authors have adequately addressed your comments raised in a previous round of review and you feel that this manuscript is now acceptable for publication, you may indicate that here to bypass the “Comments to the Author” section, enter your conflict of interest statement in the “Confidential to Editor” section, and submit your "Accept" recommendation.

Reviewer #1: All comments have been addressed

Reviewer #3: All comments have been addressed

Reviewer #4: All comments have been addressed

Reviewer #5: All comments have been addressed

2. Does this manuscript meet PLOS Digital Health’s publication criteria ? Is the manuscript technically sound, and do the data support the conclusions? The manuscript must describe methodologically and ethically rigorous research with conclusions that are appropriately drawn based on the data presented.

Reviewer #1: Yes

Reviewer #3: Yes

Reviewer #4: (No Response)

Reviewer #5: Yes

3. Has the statistical analysis been performed appropriately and rigorously?

Reviewer #1: Yes

Reviewer #3: Yes

Reviewer #4: (No Response)

Reviewer #5: Yes

4. Have the authors made all data underlying the findings in their manuscript fully available (please refer to the Data Availability Statement at the start of the manuscript PDF file)?

Reviewer #1: Yes

Reviewer #3: No

Reviewer #4: (No Response)

Reviewer #5: Yes

5. Is the manuscript presented in an intelligible fashion and written in standard English?

PLOS Digital Health does not copyedit accepted manuscripts, so the language in submitted articles must be clear, correct, and unambiguous. Any typographical or grammatical errors should be corrected at revision, so please note any specific errors here.

Reviewer #1: Yes

Reviewer #3: Yes

Reviewer #4: (No Response)

Reviewer #5: Yes

6. Review Comments to the Author

Please use the space provided to explain your answers to the questions above. You may also include additional comments for the author, including concerns about dual publication, research ethics, or publication ethics. (Please upload your review as an attachment if it exceeds 20,000 characters)

Reviewer #1: No further changes recommended.

Reviewer #3: All my comments were appropriately addressed. The manuscript reads well now.

Reviewer #4: After reviewing the changes made by the authors, I can see that they have attended to the revisions appropriately and the manuscript seems acceptable to be published within this form.

Reviewer #5: Dear Authors,

I appreciate the efforts you made with revising your manuscript and your comments back to my inquiries. As mentioned, you are researching a valuable and needed area of support that is adolescent mothers. As they are at high risk for subjugation to stigmas and discrimination, they are in need of a community or a place where they feel they belong and are accepted. Your research opens the doors for future research and further understanding of their needs and intrinsic motivations, and eventually how to change their behaviors and practices from knowledge and skill set acquisition.

All the best to you,

Reviewer #5

7. PLOS authors have the option to publish the peer review history of their article (what does this mean? ). If published, this will include your full peer review and any attached files.

**Do you want your identity to be public for this peer review?** For information about this choice, including consent withdrawal, please see our Privacy Policy .

Reviewer #1: No

Reviewer #3: No

Reviewer #4: None

Reviewer #5: No
